# Emissions reduction and pricing of supply chain under cap-and-trade and subsidy mechanisms

**Wenqing Miao[1], Guohua Zhu[2]\*, Bingliang Shen[3], Demin Kong[4]**

**1** School of Business Administration, Shanghai University of Finance and Economics Zhejiang College, Jinhua, Zhejiang, China, **2** School of Marxism, Shanghai University of Finance and Economics Zhejiang College, Jinhua, Zhejiang, China, **3** Department of Basic Education, Shanghai University of Finance and Economics Zhejiang College, Jinhua, Zhejiang, China, **4** College of Humanities, Shanghai University of Finance and Economics shanghai, Shanghai, China

\* Z2011106@shufe-zj.edu.cn

**Data Availability Statement:** All relevant data are within the paper and its Supporting Information files.

## Abstract

This paper explored how the government provides low-carbon subsidies for the manufacturers, retailers, and consumers in a secondary supply chain under cap-and-trade scheme. We calculated the best prices, emissions reductions, and the demands for common and low-carbon products when subsidizing each of the abovementioned market players. In particular, a comparative analysis of their equilibrium outcomes was made thereafter. The MATLAB simulation found that the optimal emissions reductions under the three subsidy modes were even and positively correlated to low-carbon subsidies, which, however, negatively correlated to the prices of both product types. Higher subsidies drove up demand for low-carbon products and dragged down that for common goods. But the prices of these products maintained the highest levels when consumers were subsidized; demand for common products was greater when subsidies went to retailers than to manufacturers or consumers, consequently generating the largest emissions and highest profits. When the subsidies were greater than $\frac{2p_cBJ+4p_c^2K+12k\varepsilon+4p_c^2e_1J-18e_1\varepsilon J}{48p_c\varepsilon}$, all three subsidy modes saw a drop in total carbon emissions. That being so, the government should offer proper subsidies before seeing energy-saving progress.

## 1. Introduction

Global economic growth has resulted in increased greenhouse gas emissions and worsened the environment across the world. China, as the world's second-largest economy, has stayed committed to environmental protection. In order to control carbon emissions, China has taken tough measures on carbon emissions, put forward cap-and-trade policy and began to use market mechanism to reduce emissions [1]. Since 2013, the country has piloted seven carbon emissions trading areas to curb emissions, and the total trading volume of carbon allowances has exceeded 180 million tons. The building of the national carbon market began with the promulgation of the *National Carbon Emissions Trading Market Construction Plan (Power Generation*

**Funding:** This research was supported by the Exploration Project of Natural Science Foundation of Zhejiang Province (LY20A010003) and the Development Foundation Project of Shanghai University of Finance and Economics Zhejiang College (2020GR003).

**Competing interests:** The authors have declared that no competing interests exist.

*Industry)* by the Chinese government in 2017, which saw the carbon emissions trading program as a critical measure for energy conservation and emissions reduction in China and beyond. The program is a scheme of buying and selling the right to pollute, which, to a certain extent, conserve energy and reduce emissions [2]. The research on carbon trading is based on manufacturers reducing carbon emissions per unit product through technological innovation [2], and analyzes the impact of low-carbon mechanism on emission reduction degree [3], low-carbon publicity, low-carbon emission reduction and profit [4]. However, by analyzing the green VMI Model under the carbon trading mechanism, Jiang [5] found that the introduction of carbon trading mechanism will increase the total cost of the supply chain to a certain extent.

Since the carbon trading mechanism does not apply to all situations, shortly afterward, China issued the *Interim Measures on the Administration of the Subsidy for Energy Conservation and Emissions Reduction*, allowing the government to pay a subsidy to low-carbon products and practices that save energy and curb emissions. The existing studies on low-carbon subsidies in the supply chain mainly focus on the analysis of the form of cooperation contract between retailers and suppliers and the effect of emission reduction [6–8]. In the analysis of government subsidy policies, these documents consider the consumer factors, focusing on single products, but less on multiple products. Single low-carbon subsidy mechanism can't fully meet the requirements of China's carbon emission reduction. Therefore, China proposes to combine the two mechanisms to reduce emissions [9, 10], such as the combination of carbon tax and low-carbon subsidies, cap-and-trade and low-carbon subsidies.

With the cap-and-trade and subsidy mechanisms in place, businesses will be further motivated to promote innovation, conserve energy, and reduce emissions, thus contributing to the growth of China's low-carbon economy. The low-carbon subsidy for supply chain node enterprises under the carbon trading mechanism is discussed in this paper. The cap-and-trade and low-carbon subsidy mechanism are combined to analyze their effects, makes up for the shortcomings of using cap-and-trade and low-carbon subsidy mechanism respectively in the existing literature, and is more in line with China's national conditions. Most of the existing literature only considers one product. But it is impossible for enterprises to produce only one product in actual production. This paper studies enterprises that produce common products and low-carbon products at the same time, which will expand the current supply chain cooperative emission reduction theory based on Stackelberg game. The research results of this paper have practical guiding significance for China's supply chain energy conservation and emission reduction, low-carbon economic construction and realizing the goal of carbon peak and carbon neutralization. The task of energy conservation and emission reduction of supply chain is arduous in China, and the research and application of relevant government policies are not mature. This study will provide some reference for government policy-making.

## 2. Literature review

The literature related to this paper mainly includes two aspects: cap-and-trade mechanism and low-carbon subsidy mechanism.

### 2.1 Cap-and-trade mechanism

Despite being a pilot program, the national cap-and-trade scheme has been studied by large numbers of researchers. By building a hyper network model based on the carbon market with a three-level supply chain, Ma et al. [11]. assessed the profits of parties in the supply chain with or without emissions trading, and called on the government to tighten the regulation of heavy polluters while relaxing oversight over small emitters. Under the cap-and-trade regulation, Yu et al. [12] examined different information sharing formats of an incumbent retailer with

private demand information and an uninformed entrant retailer, and considered how they could affect the manufacturer's capability to abate carbon emissions. Dong et al. [13] examined how the manufacturer, under the cap-and-trade scheme, affected the optimal decision in the whole supply chain through sustainability investment. Yenipazarli [14], Xu [15], and Miao [16] investigated the optimal decision on production, remanufacturing and pricing of a manufacturer under the emissions trading mechanism, and found that cap-and-trade regulation facilitated the production and sales of remanufactured products. All the available literature [17–20] studied the carbon trading scheme from the perspectives of the supply chain, the government, manufacturers and retailers, and delivered solid and fruitful results.

## 2.2 Low-carbon subsidy mechanism

With respect to research on low-carbon subsidies, a greater focus has been on how the subsidies impact product pricing and emissions reduction across the supply chain. For instance, Li et al. [21] considered the role of low-carbon subsidies in a closed-loop supply chain before concluding that the price for recycling the worn-out products could contribute to reducing carbon emissions only with appropriate low-carbon subsidies. Supriya Mitra et al. [22] investigated the effect of different government subsidy methods on product recycling and remanufacturing in an environment where remanufactured goods compete with new products. By examining the impact of emissions reduction on the optimal wholesale and direct sales prices when the government pays low-carbon subsidies to manufacturers, Che et al. [23] found that these prices, together with the retail price, went up as emissions reductions increased. Moreover, with low-carbon subsidies, how supply chain entities compete and cooperate also had a significant effect on the decision-making of the supply chain on pricing and production. Zhu et al. [24] and Hu et al. [25] studied the role of low-carbon subsidies in the decision-making of two competing manufacturers on production and pricing. In addition, diverse forms of subsidies have also been a research emphasis. Through setting a subsidy threshold, Li et al. [26] assessed the impact of subsidies on the pricing decisions of manufacturers and found that government incentives rendered the supply chain more eco-friendly. Wu et al. [27] investigated the cost-sharing and emissions reductions of a manufacturer and a retailer in a Nash game considering government subsidies. Based on four different governmental subsidy strategies, Xu et al. [28] proposed a Stackelberg game model that showed subsidizing both retailers and manufacturers leads to environmental protection and economic development. Meantime, how the government and businesses behave in a game involving low-carbon subsidies has also been preferred by scholars. For example, Wang et al. [29] considered the behavior concerning rules on low-carbon technology innovation of the government and companies in a dynamic game as the companies push for such technology at different stages. With an evolutionary game model, Chen et al. [30] examined the optimal carbon tax and subsidy mechanisms and production decision-making for the government and enterprises under three models: dynamic taxes and static subsidies, static taxes and dynamic subsidies, and dynamic taxes and dynamic subsidies. Research efforts on low-carbon subsidies have centered on the recycling of used products, inter-company competition, subsidy methods, and the game between the government and businesses [31–34]. Still, in most cases, only one product was covered.

These previous studies, which profile energy conservation and emissions reduction in the supply chain from the perspectives of the cap-and-trade and subsidy mechanisms, have laid solid groundwork for our research. While most of the literature on low-carbon subsidy mechanisms is about subsidizing manufacturers, this paper, based on the above research efforts, combined the emissions trading scheme with the mechanism for low-carbon subsidies. We aimed to assess the impact of low-carbon subsidies on the wholesale and retail prices, profits, carbon

emissions reductions per unit of product, and total emissions of two different products when the government provides low-carbon subsidies to manufacturers, retailers and consumers, respectively.

## 3. Research assumptions

The cap-and-trade program has been piloted in many places in China and will soon be extended nationwide. Meantime, the government is promoting low-carbon subsidies in home appliances, new energy vehicles, electricity, and other industries. Along with the program, low-carbon subsidies will underlie the development of China's low-carbon economy and influence the pattern of its economic growth in the long run. At this point, there are three types of low-carbon subsidies: 1) Subsidies to manufacturers. As the bedrock of energy conservation and emissions reduction in the supply chain, manufacturers need to invest heavily in technological innovation, which can be compensated by government subsidies. This constitutes the most common subsidy method. For instance, the central government pays subsidies to the makers of new energy vehicles. 2) Subsidies to retailers. Due to information asymmetry, carbon emissions are not familiar to most consumers, meaning retailers can help consumers better understand low-carbon products through marketing and promotion. Low-carbon subsidies are commonly used to offset the costs arising from retailers' promotion endeavors. For instance, China provides subsidies for newly-established retailing stores, green catering, and distribution enterprises. The approach, however, is only adopted in a few industries. 3) Subsidies to consumers. Those who receive a subsidy upon the purchase of low-carbon products see immediate benefits. The approach encourages consumers to buy and thus, stimulates demand. An example is energy conservation subsidies to home appliances. Within the context, the study considered a cap-and-trade scheme for the secondary supply chain that involved manufacturers and retailers, and investigated how supply chain pricing and emissions reduction were affected as the government offered low-carbon subsidies to manufacturers, retailers, and consumers, respectively.

To simplify the research, we put forth assumptions as follows.

1. Assume the market capacity of a specific product is 1, and its sales volume stands at $Q = 1 - P_1$ when the company only produces the common product as it has not upgraded its technologies. Upon technological transformation, it is capable of manufacturing both common and low-carbon products on a competitive basis. No matter how many kinds of products the enterprise produces, the total demand of consumers for enterprise products remains unchanged. When the company produces common products and low-carbon products, the sales volume of common products depends on the price gap between common products and low-carbon products, so $Q_1 = p_2 - p_1$. Then the sales volume of low-carbon goods is $Q_2 = Q - Q_1 = 1 - p_2$. Consumers have low-carbon preference. When the price of common products is equal to that of low-carbon products, they will give priority to low-carbon products, the sales volume of common products stands at 0, whilst that of low-carbon products is $Q_2 = Q$.

2. Assume the government sets the carbon cap $E_g$ for manufacturers, then carbon allowances can be viewed as tradable resources. Manufacturers can buy these rights on the carbon market when their emissions surpass the cap or sell their surplus allowances, both at the price of $p_c$.

3. Assume the carbon emissions of manufacturers in producing per unit of common product are $e_1$, and for businesses capable of producing low-carbon goods upon technological change, their emissions in the making per unit of low-carbon product are $e_2$. Because the

carbon emission of low-carbon product is less than that of ordinary product, the resulting emissions reductions stand at $e = e_1 - e_2 (0 < e < e_1)$. The input cost of emission reduction technology is an increasing convex function of emission reduction per unit product, the cost of technological transformation is $\frac{1}{2}\varepsilon e^2$.

All the symbols and descriptions in the model are listed in Table 1.

## 4. Modeling

### 4.1. Subsidies to manufacturers (M-mode)

The government pays the subsidy $s$ to manufacturers for producing one low-carbon product. At this point, the profit of manufacturers is determined by subtracting the costs of technological transformation and carbon trading from the profit that both common and low-carbon products generate (including low-carbon subsidies). Thus, its function is expressed as:

$$\pi_m^M = (w_1 - c_1)Q_1 + (w_2 - c_2 + s)Q_2 - \frac{1}{2}\varepsilon e^2 - p_c\left[e_1 Q_1 + (e_1 - e)Q_2 - E_g\right]$$

$(w_1 - c_1)Q_1$ represents the common product revenue of manufacturers; $(w_2 - c_2 + s)Q_2$ is the low-carbon product sales revenue of manufacturers upon the receipt of government subsidies; $\frac{1}{2}\varepsilon e^2$ indicates the cost of harnessing low-carbon technology; $p_c[e_1 Q_1 + (e_1 - e)Q_2 - E_g]$ denotes manufacturers' earnings or expenses from selling or buying carbon allowances.

The profit of retailers is calculated by: $\pi_r^M = (p_1 - w_1)Q_1 + (p_2 - w_2)Q_2$

Through formula 1 and formula 2, we obtain theorem 1 by backward induction approach.

**Theorem 1**: Under M-mode, the equilibrium results are as follows: $w_1^M = \frac{1 + c_1 + p_c e_1}{2}$, $w_2^M = \frac{1 + c_2 - s + p_c e_1 - p_c e}{2}$, $p_1^M = \frac{4 + c_1 + c_2 + 2p_c e_1 - p_c e - s}{6}$, $p_2^M = \frac{5 - c_1 + 2c_2 + p_c e_1 - 2p_c e - 2s}{6}$, $Q_1^M = \frac{1 - 2c_1 + c_2 - p_c e_1 - p_c e - s}{6}$, $Q_2^M = \frac{1 + c_1 - 2c_2 - p_c e_1 + 2p_c e + 2s}{6}$, $e^M = \frac{p_c^2 e_1 + 2c_2 p_c - c_1 p_c - p_c - 2p_c s}{2p_c^2 - 6\varepsilon}$.

**Proof:** Please see Appendix A in S1 Appendix.

To ensure an efficient supply chain, we assume $2\varepsilon < p_c^2 < 3\varepsilon$, then $p_c^2 e_1 + 2p_c c_2 - p_c c_1 - p_c - 2p_c s < 0$, meaning low-carbon subsidies help increase the efficiency of the supply chain only when the carbon trading price set by the government needs to be within a specific range. Thus, we let $2p_c^2 - 6\varepsilon = J$ and $p_c^2 e_1 + 2c_2 p_c - c_1 p_c - p_c = K$.

### 4.2. Subsidies to retailers (R-mode)

When retailers are the ones to receive government subsidies, each low-carbon product sold by retailers is subsidized with $s$. Then, the function of manufacturers' profits is:

$$\pi_m^R = (w_1 - c_1)Q_1 + (w_2 - c_2)Q_2 - \frac{1}{2}\varepsilon e^2 - p_c\left[e_1 Q_1 + (e_1 - e)Q_2 - E_g\right]$$

**Table 1.  List of symbols and descriptions in the model.**

| Symbols | Descriptions | Symbols | Descriptions |
|---|---|---|---|
| $s$ | Government subsidies per unit of low-carbon product | $w_1$ | Common product wholesale price |
| $E_g$ | Government carbon allowances for manufacturers | $w_2$ | Low-carbon product wholesale price |
| $E$ | Total carbon emissions of manufacturers | $p_1$ | Common product retail price |
| $e_1$ | Carbon emissions per unit of common product | $p_2$ | Low-carbon product retail price |
| $e_2$ | Carbon emissions per unit of low-carbon product ($e_2 < e_1$) | $Q_1$ | Market demand for common products ($0 \leq Q_1 \leq 1$) |
| $e$ | Carbon emissions reductions per unit of low-carbon product ($0 < e < e_1$) | $Q_2$ | Market demand for low-carbon products ($0 \leq Q_2 \leq 1$) |
| $\varepsilon$ | Coefficient of investment costs | $p_c$ | Carbon trading price |

The retailers' profit consists of the profits on common and low-carbon products, and low-carbon subsidies. Its function is expressed as $\pi_r^R = (p_1 - w_1)Q_1 + (p_2 - w_2 + s)Q_2$.

**Theorem 2**: Under R-mode, the equilibrium results are as follows: $w_1^R = \frac{1+c_1+p_c e_1}{2} - \frac{2s}{3}$, $w_2^R = \frac{1+c_2+p_c e_1 - p_c e}{2} + \frac{s}{6}$, $p_1^R = \frac{4+c_1+c_2+2p_c e_1 - p_c e - 3s}{6}$, $p_2^R = \frac{5-c_1+2c_2+p_c e_1 - 2p_c e - 2s}{6}$, $Q_1^R = \frac{1-2c_1+c_2-p_c e_1 - p_c e + s}{6}$, $Q_2^R = \frac{1+c_1-2c_2-p_c e_1+2p_c e+2s}{6}$, $e^R = \frac{p_c^2 e_1 + 2p_c c_2 - p_c c_1 - p_c - 2p_c s}{2p_c^2 - 6\varepsilon}$.

**Proof:** Please see Appendix B in S1 Appendix.

## 4.3. Subsidies to consumers (C-mode)

It is commonly seen that consumers buying low-power household appliances would receive a subsidy from the government. We assume consumers can be subsidized with *s* for purchasing a low-carbon product. Given that low-carbon subsidies have a direct impact on the demand of consumers for both common and green goods, we, thus, can have the function indicating demand for low-carbon products: $Q_2 = 1-(p_2-s) = 1-p_2+s$, and the function denoting common product demand: $Q_1 = p_2-s-p_1$. Since the functions of profits of manufacturers and retailers exclude *s*, the function of manufacturers' profits is:

$$\pi_m^C = (w_1 - c_1)Q_1 + (w_2 - c_2)Q_2 - \frac{1}{2}\varepsilon e^2 - p_c\left[e_1 Q_1 + (e_1 - e)Q_2 - E_g\right]$$

Under such a model, the function of retailers' profit can be expressed as:
$\pi_r^C = (p_1 - w_1)Q_1 + (p_2 - w_2)Q_2$

**Theorem 3**: Under C-mode, the equilibrium results are as follows: $w_1^C = \frac{1+c_1+p_c e_1}{2}$, $w_2^C = \frac{1+c_2+p_c e_1 - p_c e+s}{2}$, $p_1^C = \frac{4+c_1+c_2+2p_c e_1 - p_c e - s}{6}$, $p_2^C = \frac{5-c_1+2c_2+p_c e_1 - 2p_c e+4s}{6}$, $Q_1^C = \frac{1-2c_1+c_2-p_c e_1 - p_c e - s}{6}$, $Q_2^C = \frac{1+c_1-2c_2-p_c e_1+2p_c e+2s}{6}$, $e^C = \frac{p_c^2 e_1 + 2p_c c_2 - p_c c_1 - p_c - 2p_c s}{2p_c^2 - 6\varepsilon}$.

**Proof:** Please see Appendix C in S1 Appendix.

## 4.4. Comparative analysis

Under the emissions trading scheme, the equilibrium outcomes from the three different subsidy modes are shown in Table 2.

After comparing the wholesale and retail prices, sales volume, optimal carbon emissions reductions, and total carbon emissions under the three models, we reach the following inferences.

**Inference 1:** The optimal carbon emissions reductions per unit of low-carbon product under the three modes are equal, and are negatively correlated with the investment cost coefficient and positively correlated with low-carbon subsidies.

Proof: Please see Appendix D in S1 Appendix.

**Table 2. Equilibrium outcomes.**

| | M-Mode | R-Mode | C-Mode |
|---|---|---|---|
| $w_1$ | $\frac{1+c_1+p_c e_1}{2}$ | $\frac{1+c_1+p_c e_1}{2} - \frac{2s}{3}$ | $\frac{1+c_1+p_c e_1}{2}$ |
| $w_2$ | $\frac{1+c_2+p_c e_1 - p_c e - s}{2}$ | $\frac{1+c_2+p_c e_1 - p_c e}{2} + \frac{s}{6}$ | $\frac{1+c_2+p_c e_1 - p_c e+s}{2}$ |
| $p_1$ | $\frac{4+c_1+c_2+2p_c e_1 - p_c e - s}{6}$ | $\frac{4+c_1+c_2+2p_c e_1 - p_c e - 3s}{6}$ | $\frac{4+c_1+c_2+2p_c e_1 - p_c e - s}{6}$ |
| $p_2$ | $\frac{5-c_1+2c_2+p_c e_1 - 2p_c e - 2s}{6}$ | $\frac{5-c_1+2c_2+p_c e_1 - 2p_c e - 2s}{6}$ | $\frac{5-c_1+2c_2+p_c e_1 - 2p_c e+4s}{6}$ |
| $Q_1$ | $\frac{1-2c_1+c_2-p_c e_1 - p_c e - s}{6}$ | $\frac{1-2c_1+c_2-p_c e_1 - p_c e+s}{6}$ | $\frac{1-2c_1+c_2-p_c e_1 - p_c e - s}{6}$ |
| $Q_2$ | $\frac{1+c_1-2c_2-p_c e_1+2p_c e+2s}{6}$ | $\frac{1+c_1-2c_2-p_c e_1+2p_c e+2s}{6}$ | $\frac{1+c_1-2c_2-p_c e_1+2p_c e+2s}{6}$ |
| $e$ | $\frac{p_c^2 e_1 + 2p_c c_2 - p_c c_1 - p_c - 2p_c s}{2p_c^2 - 6\varepsilon}$ | $\frac{p_c^2 e_1 + 2p_c c_2 - p_c c_1 - p_c - 2p_c s}{2p_c^2 - 6\varepsilon}$ | $\frac{p_c^2 e_1 + 2p_c c_2 - p_c c_1 - p_c - 2p_c s}{2p_c^2 - 6\varepsilon}$ |

**N**o matter which link (manufacturers, retailers, or consumers) in the supply chain receives government subsidies, the optimal carbon emissions reductions per unit of low-carbon product are the same. If the government subsidizes low-carbon products, the supply chain as a whole invariably sees those incentives channeled into manufacturers' efforts to save energy and reduce emissions. That, ultimately, leads to equivalence in the optimal carbon emissions reductions under the three modes.

The negative correlation between the optimal emissions reductions per unit of low-carbon product and the investment cost coefficient means the greater the coefficient, the smaller the optimal emissions reductions. A larger investment cost coefficient leads to higher costs for manufacturers to pursue technological transformation. When the cost of technological change is fixed, the more negligible effect the change, the lower optimal emissions reductions.

The optimal carbon emissions reductions per unit of low-carbon product are positively correlated with low-carbon subsidies. Increased low-carbon subsidies manufacturers receive from the government will bring greater benefits. Businesses are in a better financial position to pursue technological change and go green in production. That, in turn, makes for a low-carbon virtuous cycle as manufacturers are further motivated to transform themselves technologically. This, given all other conditions unchanged, helps increase the optimal emissions reductions.

**Inference 2:** The wholesale price of low-carbon products under the C-mode is higher than that under the R- and M-modes; all the wholesale prices are negatively correlated with low-carbon subsidies. The retail prices of low-carbon products under the M- and R-modes are equal and lower than those under the C-mode; all the retail prices of low-carbon products are negatively correlated with low-carbon subsidies. The demands for low-carbon products under the three models are is equal, which are positively related to low-carbon subsidies.

Proof: Please see Appendix E in S1 Appendix.

Given $w_2^C > w_2^R > w_2^M$, the wholesale price of low-carbon products is higher when government subsidies are solely provided to consumers than when incentives solely go to retailers or manufacturers. Subsidies to manufacturers are profits in disguise, which allow them to increase sales volume through a decrease in the wholesale price. In this way, the wholesale price of low-carbon products is the lowest under the three subsidy models. When subsidies are paid to retailers, manufacturers' profits cannot be compensated. That being so, producers tend to raise the wholesale price, to which incentivized retailers are more open. The retail price is the highest as the government solely provides subsidies to consumers. Retailers can earn proper profits by increasing the retail price. When manufacturers follow suit by raising the wholesale price, retailers will shift the higher wholesale price to consumers. That is the reason the wholesale price is highest when subsidies solely go to consumers. There is a negative correlation between the low-carbon product wholesale price and low-carbon products, implying that low-carbon products register a lower wholesale price with higher low-carbon subsidies. Incentivized manufacturers are willing to put a lower price on their low-carbon products in a way that brings them more subsidies while driving market demand. Subsidies can directly deliver profits for manufacturers, allowing them to increase sales volume through a price cut. When subsidies are paid to retailers rather than manufacturers, the latter's expenses on the development of green technology cannot be compensated. In this context, they tend to shift the technological transformation cost to retailers by raising the wholesale price, to which incentivized retailers are more open.

Given $p_2^C > p_2^M = p_2^R$, when manufacturers and retailers are subsidized, the retail prices of low-carbon products are even, and the retail price of green goods is negatively correlated with low-carbon subsidies under the three modes. Businesses in the supply chain would increase sales of their low-carbon products by resorting to a price cut, so as to receive more subsidies,

which help generate profits. Hence, the green product retail price and low-carbon incentives are negatively correlated. When subsidies solely go to consumers, their price tolerance for low-carbon products is higher than when manufacturers or retailers are subsidized. Under the M- and R- modes, government subsidies encourage manufacturers and retailers to lower the product price. Therefore, the retail price of low-carbon products under the two modes is lower than that under the C-mode. The retail price of low-carbon products is negatively correlated with low-carbon subsidies. That indicates the higher government incentives, the lower the low-carbon product retail price. Companies in the supply chain can increase sales of their low-carbon products by resorting to a price cut, so as to receive more subsidies, which help generate profits. Hence, the low-carbon products retail price and low-carbon subsidies are negatively correlated. These subsidies form a vital part of retailers' profit, and to get more incentives, retailers will put a lower price on their low-carbon products as a way to increase sales.

Given $Q_2^M = Q_2^R = Q_2^C$, the demands for low-carbon products are the same under the three modes. No matter which party low-carbon incentives are paid to, the sales volume of low-carbon products increases by the same margin. And because these parties form a supply chain, external incentives invariably benefit it across the board. There is a positive correlation between demand for low-carbon goods and government subsidies, suggesting that government incentives create demand for low-carbon products. Higher subsidies mean more extra profits for manufacturers. That is why a drop in both retail and wholesale prices of low-carbon products is often accompanied by greater demand for these goods.

**Inference 3:** The wholesale and retail prices of common products under M- and C-modes are equal, and both are higher than those under R-mode. Nonetheless, the demands for these products under M- and C-modes are weaker than those under R-mode. The wholesale price is negatively correlated with low-carbon subsidies under R-mode. Low-carbon subsidies are all negatively correlated with the retail prices and demand for common products.

Proof: Please see Appendix F in S1 Appendix.

Given $w_1^M = w_1^C > w_1^R$, the wholesale prices of common goods under the M- and C-modes are equal, and both are higher than those under the R-mode. When government subsidies are given to manufacturers or consumers, the wholesale prices of common products are the same and not affected by the amount of these subsidies. When the government provides low-carbon incentives to retailers, there is a negative correlation between the wholesale price of common goods and these subsidies. That means the $w_1^R$ value drops as the $s$ value increases.

Considering $p_1^M = p_1^C > p_1^R$, the retail prices of common products are the same under the M- and C-modes, and higher than those under the R-mode. There is a negative correlation between the retail price of common goods and government subsidies. That indicates government incentives help drag down the price of common products for retail and that of low-carbon goods, since the two product categories are on a competitive basis.

Given $Q_1^R > Q_1^M = Q_1^C$, when manufacturers or consumers are subsidized, the demands for common products are even and weaker than those under the R-mode. The sales volume of common products is negatively correlated with government subsidies, meaning higher subsidies are followed by weaker demand for common goods. With subsidies to manufacturers, low-carbon products will be sold at a lower price, making them more popular with consumers than common products.

When the government gives low-carbon subsidies to retailers, the wholesale and retail prices of low-carbon products will decline, and ordinary products will follow the decline as similar products to obtain competitiveness. The increasing demand for low-carbon products has led to a decline in consumers' demand for ordinary products. And when given low-carbon

subsidies, retailers see a profit increase, allowing them to further drag down the common product price. Meantime, they are in a better financial position to promote and sell their low-carbon products, which, in turn, spurs common product sales. That is why the demand for common products under R-mode is higher than that under M- or C-mode. Moreover, the quantity of goods that retailers order from manufacturers is the largest under the three modes, meaning they are entitled to a greater discount or purchase products at a lower wholesale price.

**Inference 4:** With all other conditions unchanged, the total carbon emissions under M- and C-modes are the same, yet less significant than those under R-mode. The gap between total carbon emissions is positively correlated with low-carbon subsidies.

Proof: Please see Appendix G in S1 Appendix.

When all other conditions stay unchanged, we find: $E^R > E^M = E^C$. The total carbon emissions stay at the highest levels when retailers are solely subsidized, and they remain the same as low-carbon incentives go to manufacturers and retailers. A positive correlation between the difference in the total carbon emissions and low-carbon subsidies means the difference grows more prominent as more government incentives are provided. Considering that the optimal carbon emissions reductions of low-carbon goods are the same under the three modes, the carbon emissions per unit of low-carbon product are likewise even, so are the values of $Q_2$, and there is only a variation in the sales volume of common products. That justifies $E^R > E^M = E^C$. When subsidies are provided to retailers, manufacturers' total carbon emissions are the largest. It is because the sales volume of common goods under the R-mode is larger than that under M- and C-mode. Environmentally speaking, the government should avoid granting retailers low-carbon subsidies.

**Inference 5:** When $0 < s < \frac{2p_cBJ+4p_c^2K+12k\varepsilon-6e_1\varepsilon J}{48p_c\varepsilon}$, the total carbon emissions increase under the three modes; when $0 < \frac{2p_cBJ+4p_c^2K+12k\varepsilon-6e_1\varepsilon J}{48p_c\varepsilon} < s < \frac{2p_cBJ+4p_c^2K+12k\varepsilon+4p_c^2e_1J-18e_1\varepsilon J}{48p_c\varepsilon}$, the total carbon emissions fall under M-mode and rise under C-mode; when $s > \frac{2p_cBJ+4p_c^2K+12k\varepsilon+4p_c^2e_1J-18e_1\varepsilon J}{48p_c\varepsilon}$ and $\frac{2p_cBJ+4p_c^2K+12k\varepsilon-6e_1\varepsilon J}{48p_c\varepsilon} > 0$, the total carbon emissions register a drop under the three modes.

Proof: Please see Appendix H in S1 Appendix.

The functions of total carbon emissions under the three modes are shown in Fig 1.

When $0 < s < \frac{2p_cBJ+4p_c^2K+12k\varepsilon-6e_1\varepsilon J}{48p_c\varepsilon}$, no matter which party subsidies go to, the total carbon emissions invariably go up. That means the insufficiency of low-carbon subsidies results in few expenses for technological transformation and lower emissions reductions of low-carbon goods, while keeping the carbon emissions per unit of low-carbon products at relatively high levels. But as the subsidies increase, the sales volume of low-carbon products rises accordingly, thus driving up the total carbon emissions.

When $\frac{2p_cBJ+4p_c^2K+12k\varepsilon-6e_1\varepsilon J}{48p_c\varepsilon} < s < \frac{2p_cBJ+4p_c^2K+12k\varepsilon+4p_c^2e_1J-18e_1\varepsilon J}{48p_c\varepsilon}$, the total carbon emissions fall under the M- and C-modes, and rise under the R-mode. Once the growing low-carbon incentives get to a certain point, subsidies to both manufacturers and retailers can increase the emissions reductions of their low-carbon products. In this way, the total carbon emissions will shrink as the demand for low-carbon goods goes up and that for common products dwindles. When provided with subsidies, retailers are better positioned to promote their low-carbon products, which, in turn, stimulate common product sales, making the growth of demand for common goods slower than that under the M- and C-modes. Although heavy subsidies help reduce the carbon emissions of low-carbon products, the decrease in the carbon emissions of common products cannot cover the emissions growth of green goods, thus leading to a rise in the total carbon emissions.

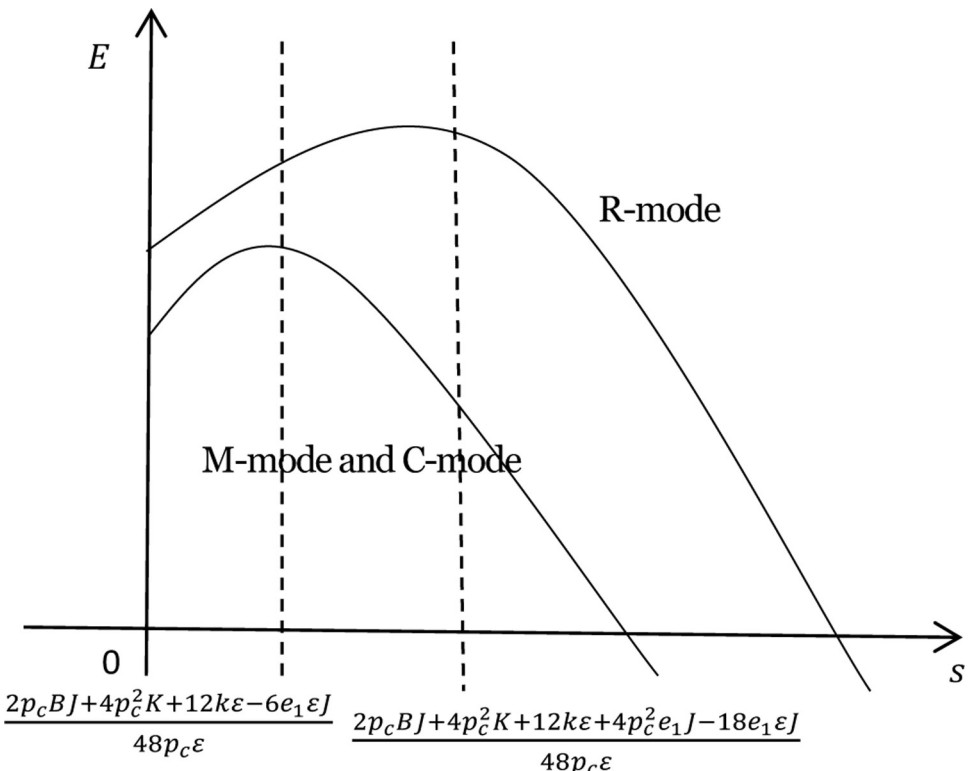

$$\frac{2p_cBJ+4p_c^2K+12k\varepsilon-6e_1\varepsilon J}{48p_c\varepsilon} \qquad \frac{2p_cBJ+4p_c^2K+12k\varepsilon+4p_c^2e_1J-18e_1\varepsilon J}{48p_c\varepsilon}$$

**Fig 1. Functions of total carbon emissions under the three modes.**

When $s > \frac{2p_cBJ+4p_c^2K+12k\varepsilon+4p_c^2e_1J-18e_1\varepsilon J}{48p_c\varepsilon}$, the total carbon emissions rise, no matter which party receives government subsidies. Heavy low-carbon incentives motivate enterprises to step up efforts on energy conservation and emissions reductions. With that, the optimal emissions reductions and carbon emissions per unit of low-carbon product plunge. Despite the increase in the demand for low-carbon products, its growth cannot cover the decline in carbon emissions per unit of low-carbon products. Furthermore, the weaker demand for common products contributes to a drop in their carbon emissions and total carbon emissions.

The context of emissions trading requires the government to firstly determine a proper carbon trading price, ensuring it satisfies $2\varepsilon < p_c^2 < 3\varepsilon$. That is how the emissions trading mechanism and low-carbon subsidies can both be given full play. Then, the most suitable subsidy receiver should be identified. Considering that when retailers are granted subsidies, the total carbon emissions are larger, and the inflection point of carbon emissions is higher compared to those under the other two modes. Hence, priority should be given to the M and C-modes. Finally, the government should find out the most appropriate number of subsidies, since a lower number of incentives would drive up the total carbon emissions, and the effects of energy conservation and emissions reduction can only be felt when the amount is within a specific range.

## 5. Calculation examples

To verify the proposed models and inferences, we took a supply chain in Shanghai as an example. The manufacturer produces common and low-carbon products that are on a competitive basis. We get the corresponding data through the investigation of the actual supply chain.

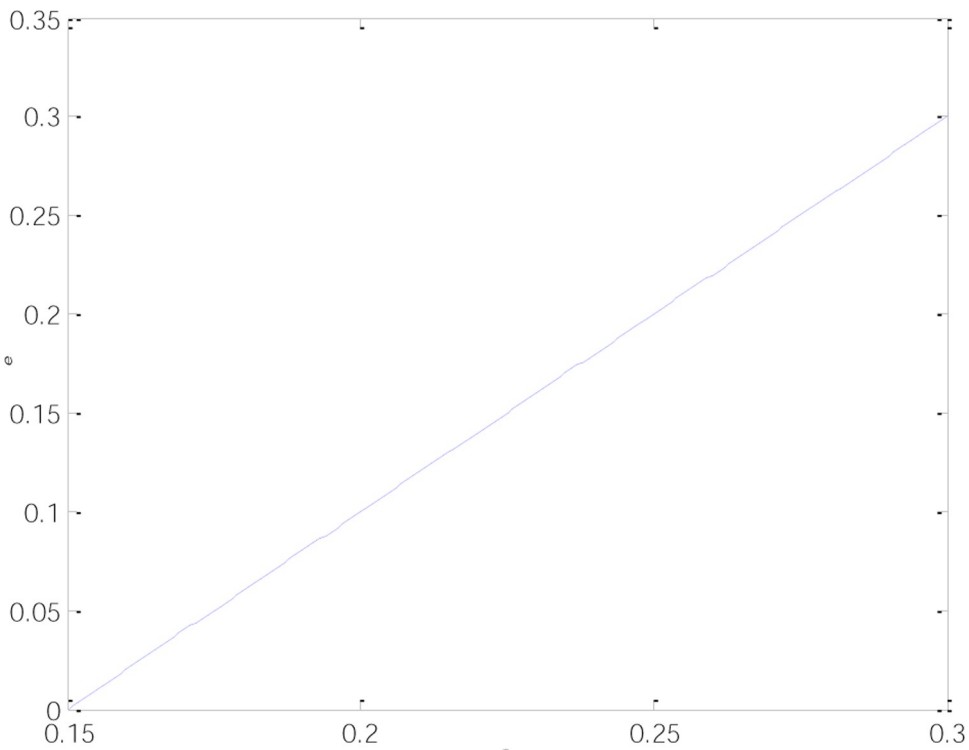

**Fig 2. Relationship between the optimal emissions reductions and low-carbon subsidies.**

Combined with the method of previous literature [35], we set the market capacity as 1, and reduce other parameters in equal proportion for the convenience of calculation. Letting $p_c = 1$, $\varepsilon = 0.4$, $c_1 = 0.2$, $c_2 = 0.5$, $e_1 = 0.5$, $E_g = 0.1$ and $s \in (0.15, 0.3)$. Upon the MATLAB simulation, we accessed the impact of different subsidy methods on carbon pricing, energy conservation, and emissions reduction.

As is shown in Fig 2, the optimal emissions reductions of low-carbon products under the three subsidy modes are the same and positively correlated with low-carbon incentives, which justifies Inference 1. Increased low-carbon subsidies can compensate for the supply chain's expenses for technological transformation and stimulate businesses to save energy and reduce emissions. Additionally, companies are better financially able to upgrade their technologies and introduce more advanced green equipment and technology for more considerable emissions reductions. The optimal emissions reductions per unit of low-carbon product grow accordingly, thus driving enterprises to step into the virtuous circle of energy savings and emissions reductions.

The relationships between the retail and wholesale prices and low-carbon subsidies are displayed in Figs 3 and 4. The results show that the wholesale and retail prices of low-carbon products are higher than those of common goods. That is because the development of low-carbon products requires heavy investment from manufacturers in improvement over the technology used to produce common goods. Afterward, the technological change cost will be shifted to the price of each low-carbon product. And we find $p_2^C > p_2^M = p_2^R$, $p_1^M = p_1^C > p_1^R$, $w_2^C > w_2^R > w_2^M$, $w_1^M = w_1^C > w_1^R$. That means under the three modes, the retail prices of common and low-carbon products are negatively correlated with low-carbon incentives, and the prices continue to fall as low-carbon subsidies increase. Despite the irrelevance to the

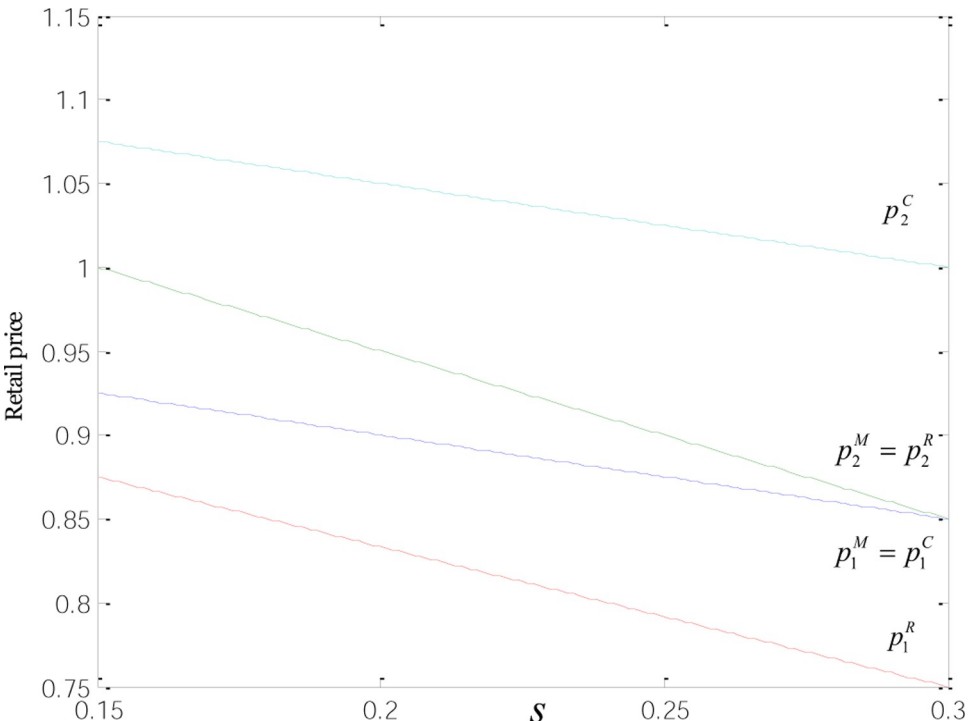

**Fig 3. Relationships between the retail price and low-carbon subsidies.**

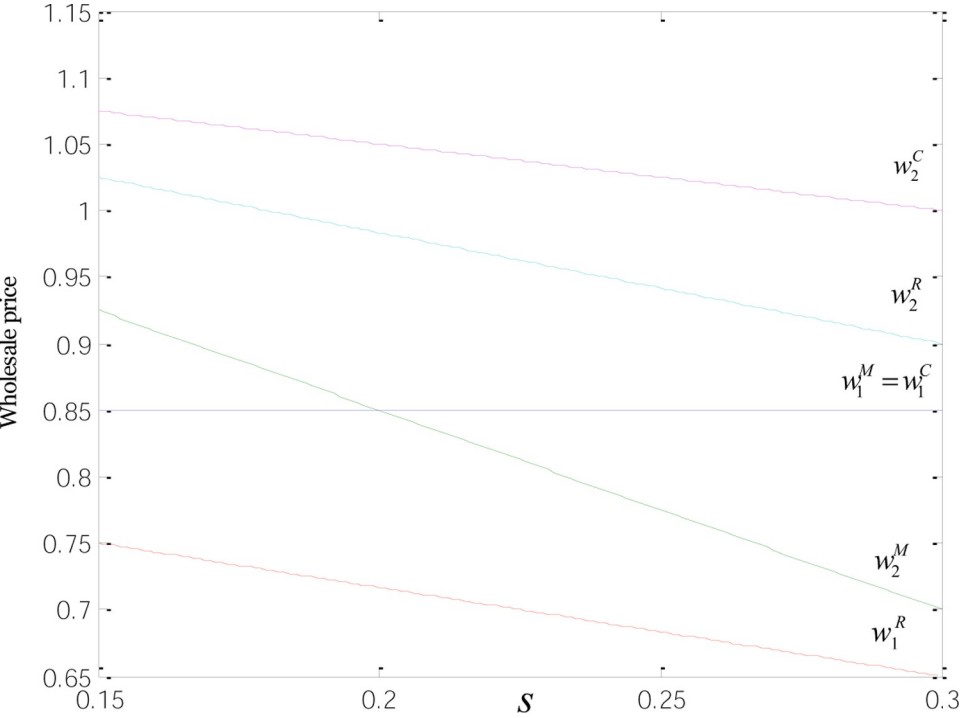

**Fig 4. Relationships between the wholesale price and low-carbon subsidies.**

wholesale prices of common goods under M- and C-modes, low-carbon subsidies are nega-tively correlated with the wholesale prices of common and low-carbon goods. When $s = 0.2$, the wholesale prices of both product types are the same under the M-mode.

When subsidies are provided to consumers rather than retailers and manufacturers, pro-ducers that have invested in technological transformation seek to raise the wholesale price of low-carbon products to shift the technology cost. Retailers likewise increase the retail price to ensure profitability. On the part of incentivized consumers, they tend to be more comfortable accepting higher prices. These have made the wholesale and retail prices of low-carbon goods under C-mode higher than those under the other two modes. When manufacturers receive subsidies, they are open to a more significant price cut since the incentives can be viewed as part of profits. That is why the wholesale price of low-carbon goods under this mode is lower than that under R- and C-mode. When retailers are solely subsidized, manufacturers tend to shift the expenses for technological change to the wholesale price of low-carbon products to ensure profitability. Retailers are more acceptable to the wholesale price of low-carbon goods when receiving subsidies, compared to the scenario when incentives solely go to manufactur-ers. Moreover, with subsidies as extra profits, retailers are more willing to offer consumers big-ger discounts, which, in turn, helps increase sales and thus obtain more incentives. At the same time, the sales volume of ordinary products in R mode is higher than that in the other two modes. So the retailers have greater bargaining power and greater quantity discount when ordering from the manufacturer. Therefore, the wholesale and retail prices of common goods, along with the retail price of low-carbon products under R-mode, are all lower than those under the other two modes, echoing the inference concerning pricing decisions in Section 3.

Fig 5 shows how the demands for common and low-carbon products relate to low-carbon subsidies. The demands for low-carbon products under the three modes are the same, echoing

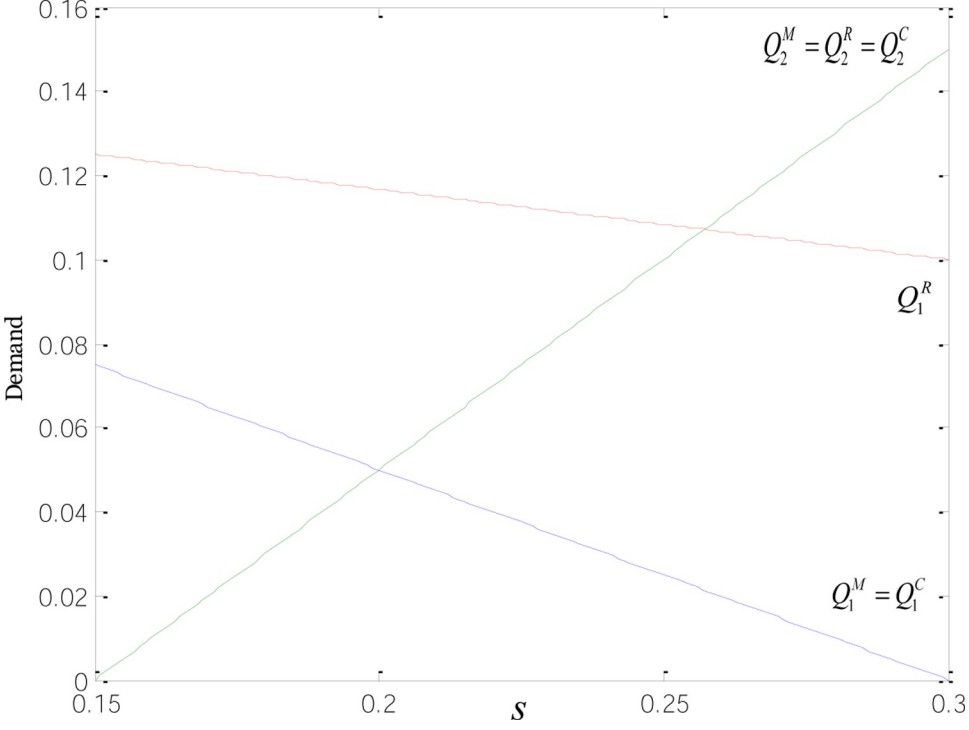

**Fig 5. Relationships between product demand and low-carbon subsidies.**

the inference regarding the demand for low-carbon goods in Inference 2, and positively correlated with low-carbon incentives. Demand for common products under the R-mode is greater than that under the M-and C-mode. When subsidies go to manufacturers or consumers, growing low-carbon subsidies are accompanied by the rising price of low-carbon products. And at this point, we see a rise in the demand for green goods and a fall in that for common products. When $s = 0.2$, the demands for both product types are even; as $s$ is approaching 0.3, the demand for common goods is nearing 0, and that for low-carbon products is at its highest levels. When retailers are the ones to receive subsidies, their stepped-up promotion efforts spur common product sales, and retailers who have more profit margins are willing to drag the retail price of common goods down to the lowest levels. That is why the demand for common products under R-mode is bigger than that under M or C-mode, consistent with Inference 3 concerning the demand for common products.

As is shown in Fig 6, there is a positive correlation between retailers' profits and low-carbon subsidies. When retailers receive subsidies, their profits are larger than when incentives solely go to manufacturers or retailers. That is because retailers, when subsidized, have larger profit margins and to snowball the incentives, they tend to promote their green goods, which, in turn, stimulates common product sales. In this way, they have more bargaining power in ordering common goods from manufacturers and usually get a lower wholesale price. That is why retailers' profit under R-mode is higher than in the other two modes.

Fig 7 describes the relationships between manufacturers' profits and low-carbon subsidies. The profit of manufacturers under the R-mode is higher than those under the M-and C-modes, since the demand for common goods is stronger when retailers are subsidized. When subsidies are offered to manufacturers or consumers, there is a positive correlation between manufacturers' profits and low-carbon subsidies, indicating that increasing subsidies are

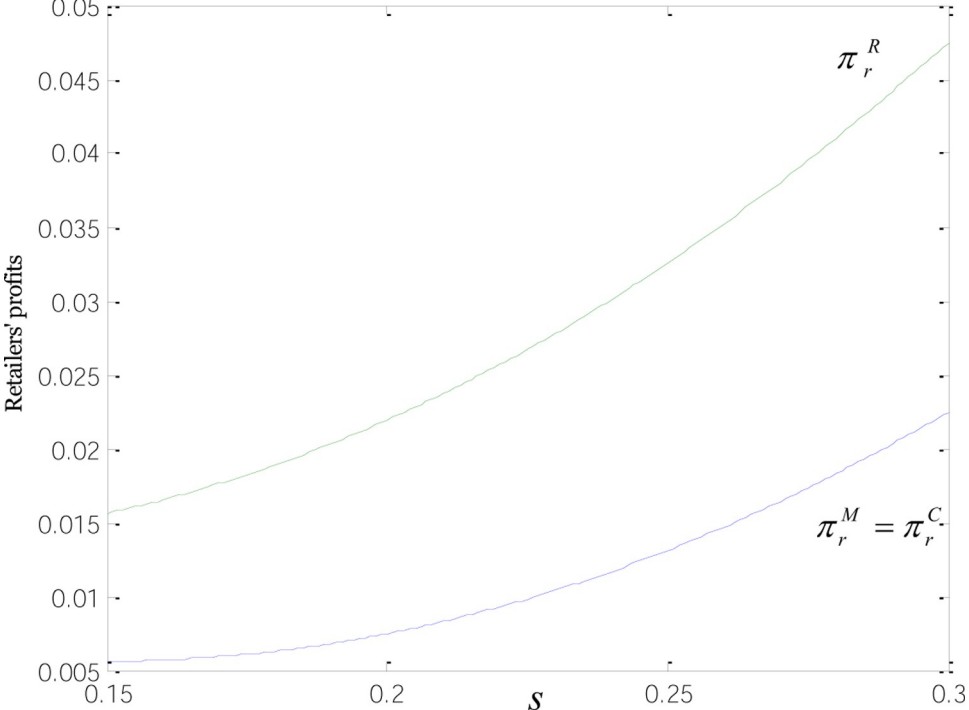

**Fig 6. Relationships between retailers' profits and low-carbon subsidies.**

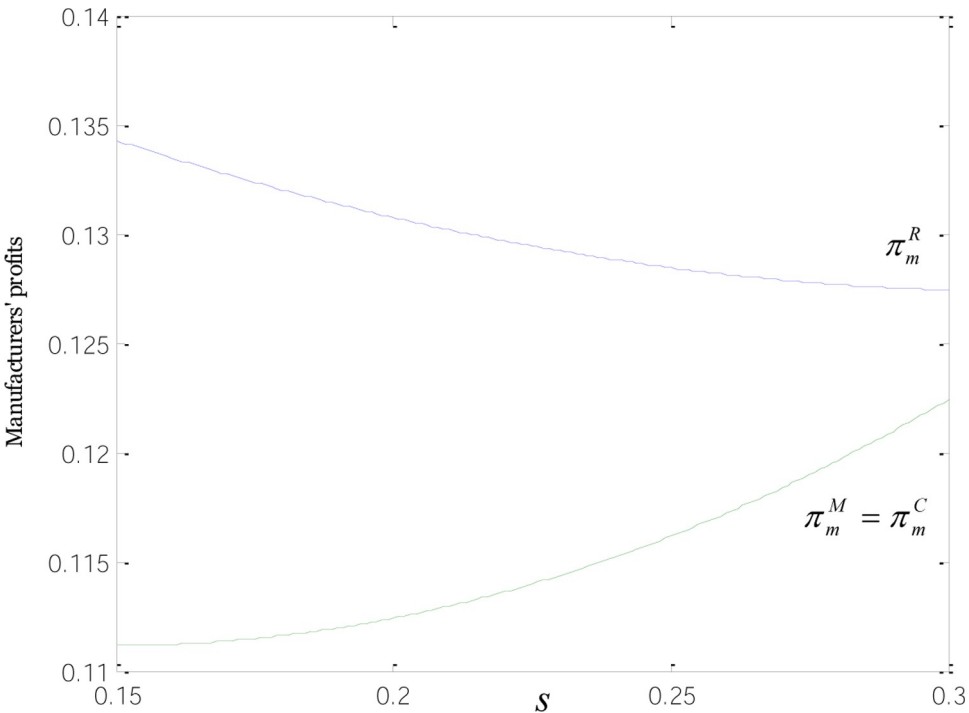

**Fig 7. Relationships between manufacturers' profits and low-carbon subsidies.**

invariably accompanied by a rise in demand for low-carbon goods and profits. But when retailers are the ones to receive incentives, manufacturers' profits are negatively correlated with low-carbon subsidies. That means as subsidies grow, retailers have a bigger say in wholesale price negotiations, leading to a fall in the demand wholesale price for common goods and in manufacturers' profits.

Fig 8 displays how total carbon emissions relate to low-carbon subsidies under the three modes. Under the R-mode, the total carbon emissions are larger than those under the M- and C-mode, and the difference in total carbon emissions is greater as low-carbon subsidies rise, which echoes Inference 10. When retailers are solely subsidized, the dropping price of common products drives up their sales volume, which is larger than under the M or C-mode. Meantime, the sales volume and emissions reductions of low-carbon goods under the R-mode are the same as under the other two modes. That causes the total carbon emissions of manufacturers to be higher under the R-mode than under the M- and C-modes. Whichever the mode it is, the total carbon emissions rise before descending, as low-carbon subsidies continue to increase, which justifies Inference 12. When subsidies are at the lower levels, the increase of incentives is followed by a slight drop in the emissions reductions of low-carbon products and the sales volume of common goods and a surge in green product sales, which contributes to an increase in total carbon emissions in the supply chain. When subsidies rise to a certain point where the supply chain is in a good financial position to apply greener technologies, the carbon emissions per unit of low-carbon product fall, followed by a decline in common product sales that helps lower total carbon emissions. Therefore, when formulating policies on low-carbon subsidies, the government should prioritize manufacturers or consumers rather than retailers, while increasing the number of subsidies. This is the way to reduce the total carbon emissions in the supply chain, making the supply chain more eco-friendly.

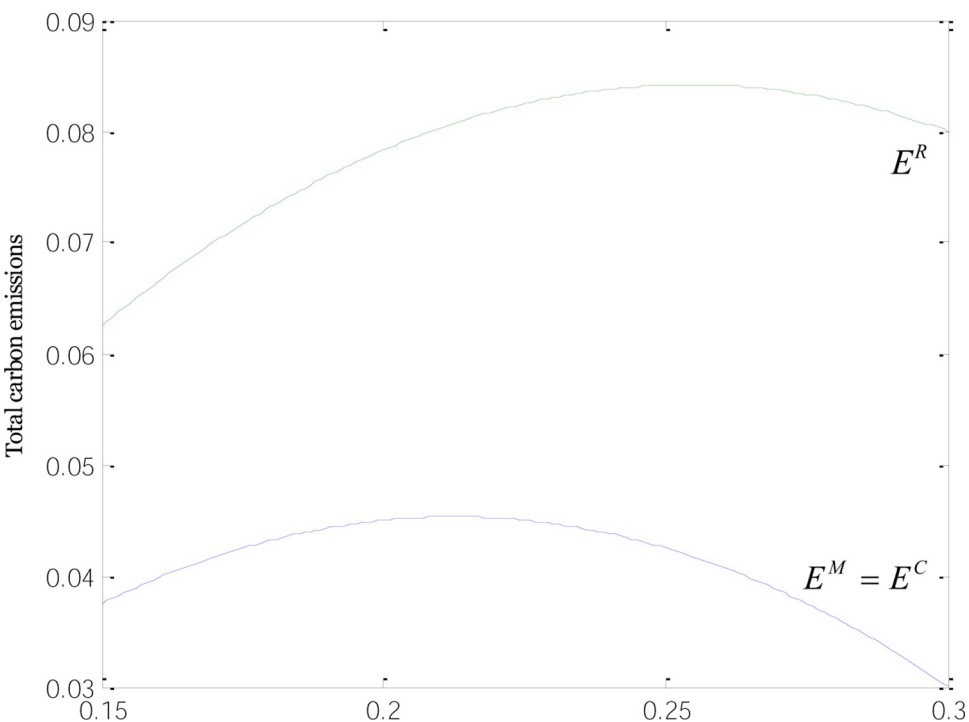

**Fig 8. Relationships between total carbon emissions and low-carbon subsidies.**

## 6. Conclusions

Under the emissions trading scheme, we provide low-carbon subsidies to manufacturers, retailers, and consumers respectively. The optimal pricing and energy conservation and emission reduction decision-making of manufacturers and retailers in a secondary supply chain under different subsidy modes are analyzed in this paper. We found the carbon price should be within an appropriate range to make the low-carbon subsidies effective, and that whichever party in the supply chain is subsidized, there is always same positive impact on energy saving and emissions reduction. High subsidies can significantly reduce the carbon emissions per unit of low-carbon product, thus encouraging the supply chain to lower the price of low-carbon products. That is followed by a rise in the sales volume of low-carbon products and a fall in that of common goods. Under the three subsidy modes, retailers invariably see a profit increase. When retailers are subsidized, the total carbon emissions and the profits of manufacturers and retailers are higher than those in other two modes. Therefore, in the implementation of subsidy policies, the government should avoid subsidizing retailers. The government should subsidize manufacturers or consumers. Considering consumers' acceptance of low-carbon products, the best model is to subsidize manufacturers, because the retail price of low-carbon products in M-mode is lower than that in C-mode, which is conducive to the future sales of low-carbon products. Besides, the subsidies should be elevated to the point where the profits of manufacturers and retailers are increasing, while the total carbon emissions are falling. It enables to motivate enterprises to save energy and reduce emissions and promotes environmental protection in ways that ensure coordinated progress in economic and social terms.

## Supporting information

**S1 Appendix.**
(DOCX)

## Author Contributions

**Conceptualization:** Wenqing Miao.

**Data curation:** Bingliang Shen.

**Formal analysis:** Bingliang Shen.

**Methodology:** Bingliang Shen.

**Project administration:** Guohua Zhu, Demin Kong.

**Supervision:** Guohua Zhu, Demin Kong.

**Validation:** Guohua Zhu, Bingliang Shen.

**Writing – original draft:** Wenqing Miao.

**Writing – review & editing:** Demin Kong.

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
