## [Decision Letter · Decision Letter 0]

31 Jan 2022

PONE-D-21-37976Emissions reduction and pricing of supply chain under cap-and-trade and subsidy mechanismsPLOS ONE

Dear Dr. Zhu,

Thank you for submitting your manuscript to PLOS ONE. After careful consideration, we feel that it has merit but does not fully meet PLOS ONE’s publication criteria as it currently stands. Therefore, we invite you to submit a revised version of the manuscript that addresses the points raised during the review process. You can find more details in the attached Editor letter.

Please provide a point-by-point response to my and reviewers’ comments, detailing what changes are made, and where in the paper these changes can be found. Please submit the revised manuscript and the response letters, but NOT a manuscript where you highlight (in red) what changes are made. I find it a lot easier to follow if it is provided in response letters instead.

We look forward to receiving your revised manuscript. Please submit your revised manuscript by Mar 17 2022 11:59PM. If you will need more time, please reply to this message or contact the journal office at plosone@plos.org.

Kind regards,

Qihong Liu

Academic Editor

PLOS ONE

Journal Requirements:

 (This research was supported by the Exploration Project of Natural Science Foundation of Zhejiang Province (LY20A010003) and the Development Foundation Project of Shanghai University of Finance and Economics Zhejiang College (2020GR003).)

(This research was supported by the Exploration Project of Natural Science Foundation of Zhejiang Province (LY20A010003) and the Development Foundation Project of Shanghai University of Finance and Economics Zhejiang College (2020GR003).)

(This research was supported by the Exploration Project of Natural Science Foundation of Zhejiang Province (LY20A010003) and the Development Foundation Project of Shanghai University of Finance and Economics Zhejiang College (2020GR003).)

6. We note you have included a table to which you do not refer in the text of your manuscript. Please ensure that you refer to Table 1, and Table 2 in your text; if accepted, production will need this reference to link the reader to the Table.

Reviewers' comments:

Reviewer's Responses to Questions

**Comments to the Author**

1. Is the manuscript technically sound, and do the data support the conclusions?

Reviewer #1: Partly

2. Has the statistical analysis been performed appropriately and rigorously? 

Reviewer #1: N/A

3. Have the authors made all data underlying the findings in their manuscript fully available?

Reviewer #1: Yes

4. Is the manuscript presented in an intelligible fashion and written in standard English?

Reviewer #1: Yes

5. Review Comments to the Author

Reviewer #1: Manuscript ID: PONE-D-21-37976

The manuscript focuses on the emissions reduction and pricing of supply chain under cap-and-trade and subsidy mechanisms. It is not a new problem, so the illustration of innovation and difference between this study and existing literature is necessary.

There are some questions:

1. Could authors provide the illustration about the innovation and contributions in section 1?

2. The literature review is not sufficient, could authors provide a new section to review the related literature?

3. Could authors point out the key reference literatures, and show the potential difference between them and your work.

4. There are three assumptions, could you provide the rationality illustration?

5. Could authors provide interaction process figure in section 3?

6. In section 4, how do authors determine the parameters’ values, please show the illustration of the rationality.

7. Which mechanism is best? Could authors provide managerial insights or decision suggestions to supply chain managers to guide them in practice?

In addition, authors should check the paper carefully before re-submission in order to avoid the typos, grammar errors or format errors (format un-unification). Besides, the expression in this paper should be improved by native speakers.

6. PLOS authors have the option to publish the peer review history of their article (what does this mean?). If published, this will include your full peer review and any attached files.

Reviewer #1: No

---

## [Author Response · Author response to Decision Letter 0]

15 Mar 2022

Dear editor and reviewer:

Thank you for your valuable comments concerning our manuscript entitled” Emissions reduction and pricing of supply chain under cap-and-trade and subsidy mechanisms”(ID: PONE-D-21-37976).Those comments are all valuable and very helpful for revising and improving our paper, as well as the important guiding significance to our researches. Thank the experts and editors for their positive comments on this manuscript. At the same time, we have studied comments carefully and have made correction in detail according to the requirements of the reply. Revised portion are marked in red in the paper. The main correction in the paper and the responds to the comments are as flowing:

Response to editor

Response to comment 1: (On p. 3, row 103, you present these as “hypotheses”. Hypotheses are things which you later test whether they hold or not. Rather, the correct term in your setting seems “assumptions”. You are merely imposing assumptions for the model.)

Response: Thank you for your good advice. We are very sorry for our incorrect writing. We have changed "hypotheses" into "assumptions" on p.3, row 115 and p.4, row 134.

Response to comment 2: (row 107, the demand for low-carbon good is Q2=1-p2. I understand that this is probably needed for the analysis to be tractable. But this is such a strong assumption, and may drive your qualitative results, you should spend some time motivating/discussing this assumption.)

Response: Thank you for your good advice. We are very sorry for we haven't described this assumption clearly. We have rewritten this part as “Assume the market capacity of a specific product is 1, and its sales volume stands at Q=1-P_1 when the company only produces the common product as it has not upgraded its technologies. Upon technological transformation, it is capable of manufacturing both common and low-carbon products on a competitive basis. No matter how many kinds of products the enterprise produces, the total demand of consumers for enterprise products remains unchanged. When the company produces common products and low-carbon products, the sales volume of common products depends on the price gap between common products and low-carbon products, so Q_1=p_2-p_1. Then the sales volume of low-carbon goods is Q_2=Q-Q_1=1-p_2.Consumers have low-carbon preference. When the price of common products is equal to that of low-carbon products, they will give priority to low-carbon products, the sales volume of common products stands at 0, whilst that of low-carbon products is Q_2=Q.”on p.4, row 137-143. 

Response to comment 3:( When comparing different modes, I am not sure whether you can or should use the same subsidy “s”. There are various reasons why you shouldn’t. For one, the subsidy is not absolute amount, but rather rate (per unit subsidy). Since the amount of output differs across modes, the total subsidy amounts will differ as well. I would thus suggest using different notations: s_M, s_R and s_C. The question then is how to compare across modes. One possibility is to restrict total subsidy amount to be the same across modes. The other is to add a stage where the government chooses “s” under each mode to maximize its objective function (social surplus?))

Response: Thank you for your good advice. However, we believe that it will be more meaningful to compare the profits and total carbon emissions of supply chain members under three different modes with the same “s”. “s” is per unit subsidy, which will affect the sales volume. Indeed, it will lead to different total subsidies under different modes, but this is not the goal of government decision-making. The purpose of the government's low-carbon subsidy mechanisms is to reduce the total carbon emissions of enterprises, so the government will choose the mode with the lowest total carbon emissions, i.e. mode M or mode C. When s>(2p_c BJ+4p_c^2 K+12kε+4p_c^2 e_1 J-18e_1 εJ)/(48p_c ε), the total carbon emissions will decrease. So the government should increase the subsidy. This part is shown on p.10-11, row 307-342. The total carbon emissions are also shown in Figure 1 and Figure 8. The conclusion of the manuscript has also put forward suggestions to the government that subsidies to retailers should be avoided and the subsidies should be increased to reduce total carbon emissions. 

Response to comment 4: (Presentation of results)

Response: Thank you for your good advice. We have made correction according to your suggestion. We have put the detailed derivation steps in the Appendix, and only the equilibrium results are presented in the text. We have combined 12 inferences into 5 inferences, and leave the detailed proofs in the Appendix. Inference 1 is about the optimal carbon emission reduction. Inference 2 is about the wholesale price, retail price and sales volume of low-carbon products. Inference 3 is about the wholesale price, retail price and sales volume of common products. Inference 4 is about total carbon emissions. Inference 5 is about the relationship between total carbon emissions and s. On p.7-9, row 206-296.

Response to comment 5: (On p. 8, row 240, did you verify that 𝑄1≥0 is satisfied in the equilibrium?)

Response: Thank you for your good advice. We are very sorry for we have lost the range of sales volume. We have added 〖0≤Q〗_1≤1 and 〖0≤Q〗_2≤1 into Table.1 on p.5.

Response to reviewer

Response to comment 1: Could authors provide the illustration about the innovation and contributions in section 1?

Response: Thank you for your good advice. We have added the innovation and contribution to Section 1. On p.2, row 53-63.

Response to comment 2: The literature review is not sufficient, could authors provide a new section to review the related literature?

Response: Thank you for your good advice. We have added references [1]-[10] and a literature review section as you suggested. On p.1-2, row 29-50, 64-67.

Response to comment 3: Could authors point out the key reference literatures, and show the potential difference between them and your work.

Response: Thank you for your good question. The key references are [2] [9] [15] [17] [28] [35]. The above references only study the emission reduction effect of a single product from the perspective of carbon trading or low-carbon subsidies. The difference of our paper is to combine cap-and-trade mechanism with subsidy mechanism, compare the different effects of different subsidy modes. And we study the carbon emission reduction of the supply chain that produces two products at the same time.

Response to comment 4: There are three assumptions, could you provide the rationality illustration?

Response: Thank you for your good advice. We are very sorry we didn't explain these three assumptions clearly. For assumption 1, we added the relevant explanation on page 4, row 138-143. Assumption 2 is the axiom of carbon trading mechanism. All relevant papers on carbon trading mechanism assume this. And we added some explanations to assumption 3 on p.4 row 149-151.

Response to comment 5: Could authors provide interaction process figure in section 3?

Response: Thank you for your good advice. We are very sorry we didn't provide an interactive process figure. However, our subsidies to the three models are independent, so there is no interactive process figure. The interaction figure between variables under each model is put in the example analysis in section 5.

Response to comment 6: In section 4, how do authors determine the parameters’ values, please show the illustration of the rationality.

Response: Thank you for your good advice. We have rewritten this part as “To verify the proposed models and inferences, we took a supply chain in Shanghai as an example. The manufacturer produces common and low-carbon products that are on a competitive basis. We get the corresponding data through the investigation of the actual supply chain. Combined with the method of previous literature [35], we set the market capacity as 1, and reduce other parameters in equal proportion for the convenience of calculation.” On p.11, row 344-348.

Response to comment 7: Which mechanism is best? Could authors provide managerial insights or decision suggestions to supply chain managers to guide them in practice?

Response: Thank you for your good advice. This paper mainly discusses the government's subsidies to manufacturers, retailers and consumers in the supply chain, and takes the total carbon emission as an index to evaluate which subsidy mode is the best. It mainly puts forward suggestions on the government's subsidy mechanism. I'm sorry we didn't make it clear which mode is the best. However, in the conclusion, we have pointed out that the government should choose to increase subsidies to a certain extent in order to reduce total carbon emissions, and the government should not subsidize retailers. When subsidies are given to manufacturers or consumers, their total carbon emissions are equal and the lowest. But we didn't make it clear. If the total carbon emissions and consumers' acceptance of low-carbon products are considered at the same time, M-mode should be selected, because the retail price of low-carbon products under M-mode is the lowest, which is more conducive to the sales of low-carbon products in the future. So we have rewritten this part. On p.15, row 441-457. 

We appreciate for your warm work earnestly, and hope that the correction will meet with approval. Once again, thank you very much for your comments and suggestions.

Yours Sincerely

Miao wenqing, Zhu guohua, Shen bingliang, Kong demin

---

## [Decision Letter · Decision Letter 1]

21 Mar 2022

Emissions reduction and pricing of supply chain under cap-and-trade and subsidy mechanisms

PONE-D-21-37976R1

Dear Dr. Zhu,

The revision was sent to the reviewer, who thinks that all his/her comments have been satisfactorily addressed. I have also checked and believe that my comments have been addressed as well. Therefore, I am pleased to inform you that your manuscript has been judged scientifically suitable for publication and will be formally accepted for publication once it meets all outstanding technical requirements.

Kind regards,

Qihong Liu

Academic Editor

PLOS ONE

Additional Editor Comments (optional):

Reviewers' comments:

Reviewer's Responses to Questions

**Comments to the Author**

1. If the authors have adequately addressed your comments raised in a previous round of review and you feel that this manuscript is now acceptable for publication, you may indicate that here to bypass the “Comments to the Author” section, enter your conflict of interest statement in the “Confidential to Editor” section, and submit your "Accept" recommendation.

Reviewer #1: All comments have been addressed

2. Is the manuscript technically sound, and do the data support the conclusions?

Reviewer #1: Yes

3. Has the statistical analysis been performed appropriately and rigorously? 

Reviewer #1: Yes

4. Have the authors made all data underlying the findings in their manuscript fully available?

Reviewer #1: Yes

5. Is the manuscript presented in an intelligible fashion and written in standard English?

Reviewer #1: Yes

6. Review Comments to the Author

Reviewer #1: (No Response)

7. PLOS authors have the option to publish the peer review history of their article (what does this mean?). If published, this will include your full peer review and any attached files.

Reviewer #1: No

---

## [Editor Report · Acceptance letter]

24 Mar 2022

PONE-D-21-37976R1 

Emissions reduction and pricing of supply chain under cap-and-trade and subsidy mechanisms 

Dear Dr. Zhu:

I'm pleased to inform you that your manuscript has been deemed suitable for publication in PLOS ONE. Congratulations! Your manuscript is now with our production department. 

Kind regards, 

on behalf of

Dr. Qihong Liu 

Academic Editor

PLOS ONE